# Bioconversion of Deoxynivalenol by Mealworm (*Tenebrio molitor*) Larvae: Implications for Feed Safety and Nutritional Value

**DOI:** 10.3390/toxins17100478

**Published:** 2025-09-25

**Authors:** Marcin Wróbel, Michał Dąbrowski, Michał Łuczyński, Krzysztof Waśkiewicz, Tadeusz Bakuła, Łukasz Nowicki, Łukasz Zielonka

**Affiliations:** 1Department of Research and Development, Chemprof, Gutkowo 54B, 11-041 Olsztyn, Poland; marcin.wrobel@chemprof.pl (M.W.); michal.luczynski@chemprof.pl (M.Ł.); krzysztof.waskiewicz@chemprof.pl (K.W.); 2Department of Veterinary Prevention and Feed Hygiene, Faculty of Veterinary Medicine, University of Warmia and Mazury in Olsztyn, Oczapowskiego 13/29, 10-718 Olsztyn, Poland; michal.dabrowski@uwm.edu.pl (M.D.); bakta@uwm.edu.pl (T.B.); 3Altium International Sp. z o.o., ul. Puławska 303, 02-785 Warszawa, Poland; lukasz.nowicki@altium.net

**Keywords:** deoxynivalenol (DON), *Tenebrio molitor*, bioconversion, insect protein, mycotoxin detoxification, amino acid profile, fatty acid composition

## Abstract

Deoxynivalenol (DON) is one of the most common trichothecene mycotoxins found in cereals, posing a significant hazard to food and feed safety. Insects, especially the yellow mealworm (*Tenebrio molitor*), offer promising alternative protein sources; however, their capacity to metabolise mycotoxins and the nutritional implications are still not fully understood. In this study, *T. molitor* larvae were reared for two weeks on diets containing DON at 663 or 913 µg/kg, and their biomass was analysed using Liquid Chromatography–Quadrupole Time-of-Flight Mass Spectrometry (LC-QTOF) for DON metabolites and free amino acids, as well as Gas Chromatography–Flame Ionization Detector (GC-FID) for fatty acid profiles. Larvae metabolised DON via multiple pathways, including sulfonation, glucuronidation, sulfation, glucosylation, and de-epoxidation, with a time- and dose-dependent shift towards glucosylation and de-epoxidation. DON exposure significantly reduced the levels of essential amino acids such as methionine, lysine, phenylalanine, and isoleucine, and lowered metabolic intermediates like aspartic and glutamic acid. Conversely, prolonged DON exposure increased linoleic acid levels in larval fat, indicating altered lipid metabolism. These findings demonstrate that *T. molitor* larvae detoxify DON but incur measurable metabolic costs, leading to changes in amino acid and fatty acid profiles. The dual effect—reduction of toxin levels and nutritional shifts—highlights both the potential and the challenges of using insects for sustainable feed production.

## 1. Introduction

The increasing global demand for sustainable protein sources has spurred efforts to identify alternative ingredients for feed and food. Among these, insects have become promising options because of their nutritious content, efficient feed conversion, and low environmental impact [1,2]. The yellow mealworm (*Tenebrio molitor*) has attracted particular interest, as its larvae are rich in proteins, essential amino acids, polyunsaturated fatty acids, and micronutrients [2,3]. Moreover, mealworms can be cultivated on low-value side streams and agricultural by-products, making them an attractive component of circular economy models [2,3]. Their approval as a novel food and feed ingredient by the European Food Safety Authority (EFSA) further confirms their safety and potential in human and animal nutrition [4]. One of the primary risks to food and feed safety is contamination with mycotoxins. These toxic fungal metabolites, mainly produced by *Fusarium*, *Aspergillus*, and *Penicillium* species, are commonly found in cereal grains and their derivatives [5,6,7,8]. It is estimated that up to 25% of global crops are contaminated with mycotoxins, with prevalence exceeding 60–80% in certain commodities, especially cereals [9,10,11]. Among these, deoxynivalenol (DON), a type B trichothecene produced by *Fusarium graminearum* and *Fusarium culmorum*, is one of the most frequently detected contaminants in wheat, maize, barley, and oats [7,8,12,13]. DON contamination affects animal performance by reducing feed intake, growth, gut health, and immunity [12,14,15]. In humans, acute exposure can cause nausea, vomiting, and abdominal pain, while long-term exposure is linked to weakened immunity and potential metabolic problems [15]. Notably, climate change scenarios forecast an increased risk of DON and other mycotoxins in Europe and worldwide, emphasising the importance of effective management strategies [14,15,16,17,18]. The toxicokinetics and metabolism of DON have been extensively studied in vertebrates. After ingestion, DON is quickly absorbed in the gastrointestinal tract and distributed to tissues [12,14]. In pigs, which are among the most sensitive livestock species, detoxification capacity is limited; DON mainly appears in plasma and urine as the parent compound or glucuronide conjugates, causing significant productivity losses even at low dietary levels [12,14]. In poultry, tolerance is higher due to faster intestinal transit and more efficient liver conjugation, with DON-3-glucuronide as the main metabolite [12,14]. In ruminants, microbial activity in the rumen provides a natural detoxification pathway via de-epoxidation to deepoxy-DON (DOM-1), a less toxic metabolite [12,14]. However, detoxification efficiency can vary depending on rumen conditions, diet, and microbial diversity [8]. Fish and aquaculture species also exhibit differing sensitivities to DON, with rainbow trout being particularly susceptible. The metabolism in these species mainly depends on liver conjugation and some microbial activity [19]. These interspecies differences highlight the importance of species-specific toxicokinetic assessments to properly evaluate DON-related risks in food-producing animals.

Insects provide a unique and relatively underexplored model for studying mycotoxin metabolism. *T. molitor* larvae have been shown to tolerate diets contaminated with DON without significant mortality or growth inhibition [20,21]. Several studies suggest that insects can biotransform DON into less toxic metabolites, possibly through cytochrome P450-mediated hydroxylation and de-epoxidation, followed by conjugation pathways such as glycosylation or sulfation [21,22,23,24]. Importantly, the insect gut microbiota may play a role similar to that of ruminal microbiota in cattle, aiding in the degradation of DON [25]. Feeding studies with *T. molitor* also demonstrated the bioconversion of other mycotoxins, including zearalenone and T-2/HT-2 toxins, further highlighting the metabolic versatility of insects [21,26,27]. Beyond detoxification, exposure to DON can alter nutrient metabolism in insects. In *T. molitor* larvae, DON intake has been linked to changes in fatty acid profiles, such as a decrease in polyunsaturated fatty acids and a shift in the n-3/n-6 ratio, along with disrupted amino acid utilisation and protein turnover [3,24]. These findings suggest that although larvae may mitigate mycotoxin toxicity, the process could entail metabolic costs that affect biomass composition and nutritional value. Such dual outcomes—detoxification ability and metabolic adjustments—highlight the importance of carefully assessing the safety and quality of insect-derived biomass.

From a practical standpoint, understanding the metabolic fate of DON in *T. molitor* is vital for two main reasons. First, it ensures the safety of insect biomass used as feed and food ingredients by determining whether DON or its metabolites remain in tissues [8,20]. Second, it highlights the potential of insects as biological tools for valorising contaminated feedstuffs, transforming otherwise unusable materials into safe and valuable sources of protein and lipids [2,17,18]. Besides their role as a novel feed ingredient in terrestrial livestock production, *T. molitor* larvae show particular promise for aquaculture systems. The aquaculture sector is increasingly challenged by sustainability issues related to traditional protein and lipid sources, particularly fishmeal and fish oil, which are predominantly derived from wild-caught forage fish [1,2]. Larvae of *T. molitor* contain 45–60% crude protein on a dry matter basis and 25–35% fat, including beneficial mono- and polyunsaturated fatty acids [2,3]. This nutritional profile makes mealworm meal and oil appealing alternatives to conventional marine ingredients. Feeding trials have shown that partially replacing fishmeal with insect-derived protein can maintain or even enhance growth performance, feed efficiency, and gut health in species like tilapia, rainbow trout, and salmonids, as long as diets are appropriately balanced to meet essential amino acid and fatty acid requirements [1]. Additionally, mealworm oil, rich in oleic and linoleic acids, can partly substitute fish oil in aquafeeds. However, supplementation with long-chain n-3 fatty acids (EPA, DHA) from other sources remains crucial [2,3]. From a sustainability perspective, incorporating *T. molitor* biomass into aquafeeds helps reduce pressure on wild fish populations, lowers the environmental impact of aquaculture, and promotes circular economy practices by recycling agricultural by-products into valuable feed ingredients [2,17,18]. Moreover, the larvae’s capacity to biotransform and potentially detoxify mycotoxins such as DON offers an additional safety benefit, decreasing the risk of toxin transfer into farmed fish and ultimately into the human food chain [8,24]. Similar carry-over phenomena have been reported in poultry, where *Fusarium* toxins were detected in tissues and eggs after dietary exposure [28]. Therefore, this study aims to explore how DON is processed in *T. molitor* larvae and to evaluate its impact on fatty acid and protein metabolism. These findings will provide insights into the safety of insect-based biomass and help develop innovative strategies for sustainable feed production and mycotoxin detoxification.

## 2. Results

### 2.1. Metabolite Profiling of Deoxynivalenol in Tenebrio molitor Larvae

The Q-TOF analysis revealed distinct profiles of deoxynivalenol (DON) metabolites in *Tenebrio molitor* larvae, varying by experimental group and exposure duration. The results are summarised in Figure 1 and Figure 2.

In positive ionisation mode (Figure 1), DON was the most abundant molecule in the larvae from both experimental groups (A and B), with notably high levels after the first week. The presence of DON-sulfonate was detected only in the high-dose group (B) during week 1, suggesting that sulfonation may serve as an inducible detoxification pathway activated under higher exposure conditions. Additionally, small amounts of DON-3-glucuronide and DON-3-glucoside were detected, indicating that conjugation processes occurred, albeit at lower levels than the parent toxin. By week two, DON levels decreased in the high-dose group (B) but remained elevated in the low-dose group (A). Meanwhile, levels of de-epoxy-DON increased in both groups, indicating the ongoing activation of reductive detoxification pathways. In the negative ionisation mode (Figure 2), conjugated metabolites dominated the metabolic profile. DON-3-glucuronide and DON-3-sulfate were consistently present in all groups, including the control, which may reflect baseline detoxification processes in the larvae. DON-3-glucoside notably increased in the high-dose group (B) during the second week, suggesting that glucosylation became a key detoxification pathway under sustained DON exposure. This finding supports the idea that glycosylation functions as a vital protective mechanism against trichothecene mycotoxins. Overall, these findings demonstrate that *T. molitor* larvae can metabolise DON through various biotransformation pathways, including sulfonation, glucuronidation, glucosylation, and de-epoxidation. The observed patterns, depending on time and dose, show that higher DON levels initially lead to increased sulfonation, while prolonged exposure shifts the metabolism towards glucosylation and reduction. Interestingly, DON concentrations were higher in groups C and A after two weeks compared to week one. This may reflect cumulative effects due to variable feed intake and slower metabolic clearance at lower exposure levels. In contrast, the high-dose group (B) exhibited reduced DON levels at week two, indicating faster activation of detoxification pathways under stronger exposure pressure.

### 2.2. Amino Acid Profile of Tenebrio molitor Larvae

The analysis of free amino acid content in *T. molitor* larvae revealed distinct effects of dietary deoxynivalenol (DON) exposure that became increasingly evident over time. Week 1: after the first week of feeding, notable differences were observed in several amino acids among the experimental groups (Table 1). The complete LC–QTOF amino-acid dataset for all groups, time points, and replicates is provided in the Appendix A.

Compared to the control group, larvae exposed to DON showed reduced levels of aspartic acid, glutamic acid, isoleucine, lysine, methionine, phenylalanine, proline, serine, and threonine (*p* < 0.05). In contrast, there were no significant changes in the levels of arginine, cysteine, glutamine, histidine, tryptophan, or tyrosine. These findings suggest that short-term DON exposure selectively decreases amino acids involved in protein synthesis and energy metabolism.

After two weeks, the differences between groups became more noticeable (Table 2).

Notable decreases were observed in larvae exposed to DON for amino acids such as aspartic acid, glutamic acid, isoleucine, lysine, methionine, phenylalanine, proline, serine, and threonine compared to the control group (*p* < 0.05). The most significant declines were seen in methionine and phenylalanine, with levels in Group B dropping by approximately 50% relative to the controls. Conversely, tryptophan levels tended to increase in DON-fed groups, although this was not statistically significant. These differences are illustrated in Figure 3, which shows representative essential and non-essential amino acids substantially affected by DON exposure.

Overall, prolonged DON exposure caused a consistent decrease in both essential and non-essential amino acids, particularly those involved in protein synthesis (methionine, lysine, and threonine) and major metabolic pathways (aspartic and glutamic acid). These findings indicate that DON metabolism in larvae may impose a metabolic burden that disturbs amino acid balance and protein turnover.

### 2.3. Fatty Acid Profile of Tenebrio molitor Larvae

The fatty acid composition of *T. molitor* larvae was examined after one and two weeks of feeding on either the control diet (C) or diets contaminated with deoxynivalenol (Group A, Group B). Oleic acid (C18:1 cis-9) and linoleic acid (C18:2 cis-9,12) were the dominant fatty acids across all treatments, with palmitic acid (C16:0) also detected. Smaller amounts of myristoleic acid (C14:1 cis-9), palmitoleic acid (C16:1 cis-9), and stearic acid (C18:0) were additionally identified.

After one week of feeding, no significant differences were observed among the groups concerning the concentrations of major fatty acids (Table 3). The complete GC-FID fatty-acid dataset (replicate-level values) is provided in the Appendix A.

Oleic acid made up about 44% of total fatty acids, linoleic acid was around 22–23%, and palmitic acid was approximately 16–17%. Stearic acid remained steady at about 4%. In the second week, the overall fatty acid profile stayed mostly the same, except for linoleic acid, which was significantly higher in larvae from the DON-fed groups compared to the control (*p* < 0.05). Specifically, linoleic acid increased from 22.60 ± 0.32 mg/g in the control group to 24.75 ± 0.73 mg/g in Group A and 24.35 ± 0.91 mg/g in Group B. A slight trend towards lower palmitic acid levels was also seen in the DON-exposed groups, although these changes were not statistically significant. Oleic acid remained the most common fatty acid (~41–42%) and was unaffected by the dietary treatment (Table 4).

Overall, the results suggest that prolonged exposure to DON may change the fatty acid profile of *T. molitor* larvae, mainly by increasing levels of linoleic acid. These changes could reflect a metabolic response linked to DON detoxification, potentially involving modifications in lipid metabolism.

## 3. Discussion

The current study shows that *Tenebrio molitor* larvae can withstand diets contaminated with DON and break down the toxin through different biotransformation pathways. Importantly, these detoxification processes are associated with clear changes in amino acid and fatty acid profiles, suggesting that detoxification is not a metabolically neutral process. Our findings offer mechanistic insights into insect resilience against mycotoxins and underscore their potential for sustainable feed production.

### 3.1. Metabolism of Deoxynivalenol in T. molitor

Several DON metabolites were identified, including DON-sulfonate, DON-3-glucuronide, DON-3-sulfate, DON-3-glucoside, and deepoxy-DON, which indicate that multiple detoxification pathways are active [14,21,24,29]. The brief detection of DON-sulfonate in the high-dose group during the first week suggests that sulfonation acts as an inducible, short-term response to acute exposure. Sulfonated trichothecene derivatives have been identified in mammals and are regarded as detoxification products with reduced activity [14]. In contrast, glucuronidation and sulfation were consistently detected across all groups, reflecting detoxification mechanisms well known in vertebrates [12,14]. The significant increase in DON-3-glucoside with prolonged exposure highlights the importance of glucosylation. In plants, DON-3-glucoside is considered a modified mycotoxin [29,30,31,32]; its detection in insects suggests that similar conjugation pathways may also operate in invertebrates. Since modified forms can be hydrolysed back to DON in the gastrointestinal tract [29], their presence has important safety implications. The detection of deepoxy-DON (DOM-1) is particularly noteworthy, as this metabolite has significantly lower toxicity [12,14], and its formation is well established in ruminants via ruminal microbiota [14]. Our findings suggest that a similar de-epoxidation process occurs in insects, probably facilitated by their gut microbes [33,34,35], consistent with other insect species where symbionts are involved in mycotoxin breakdown [36]. Both host enzymes and microbial activity likely drive the metabolic changes observed. Cytochrome P 450 monooxygenases, which are highly diverse in insects, may catalyse hydroxylation and subsequent conjugation (e.g., glucuronidation, glucosylation) [22,23], while sulfotransferases could explain the temporary sulfonation seen in the high-dose group. This integrated host–microbiota system offers a broader metabolic capacity than many vertebrate species and may account for the high tolerance of *T. molitor* larvae to DON. Similar adaptive responses have also been noted for other *Fusarium* toxins, including zearalenone and T-2/ht-2 [26,27,36,37].

### 3.2. Comparative Toxicological Perspectives Across Species

The metabolic responses of *T. molitor* resemble those of vertebrates but also display insect-specific adaptations. In pigs, one of the most sensitive species, DON metabolism is restricted to glucuronidation, which does not sufficiently reduce toxicity [12,14]. Poultry tolerate DON better due to rapid intestinal transit and liver conjugation [12,14]. Ruminants benefit from ruminal de-epoxidation to DOM-1 [12]. Fish show variable sensitivity: rainbow trout are highly susceptible [19], whereas carp mainly exhibit immune changes rather than growth suppression [38]. In contrast, *T. molitor* employs conjugation pathways similar to those of vertebrates but also activates sulfonation and glucosylation. This broader metabolic spectrum probably results from evolutionary adaptation to frequent exposure to fungal metabolites in natural environments. Such metabolic flexibility may help explain the remarkable resilience of insects compared to vertebrates [22]. These interspecies comparisons highlight the importance of insect-specific toxicokinetic assessments when evaluating feed safety.

### 3.3. Effects of DON on Amino Acid Metabolism

DON exposure caused consistent reductions in several essential and non-essential amino acids, especially methionine, lysine, phenylalanine, isoleucine, aspartic acid, and glutamic acid. These amino acids are crucial for protein synthesis and fundamental cellular functions. The reduction—up to 50% for methionine and phenylalanine and about 25% for lysine relative to controls—is nutritionally relevant. When compared to poultry and swine dietary requirements, such decreases may lower the protein quality of insect-derived biomass. While detoxification reduces the risk of DON carry-over, it comes at the cost of diminished amino acid availability. Interestingly, tryptophan levels tended to increase in DON-fed larvae, possibly suggesting a shift towards serotonin or kynurenine pathways, which may be associated with stress or immune responses. Similar shifts have been linked to DON-induced immune dysregulation in vertebrates [15].

### 3.4. Effects of DON on Fatty Acid Composition

The fatty acid profile of *T. molitor* was primarily composed of oleic and linoleic acids, consistent with previous reports [2,3]. Extended DON exposure significantly raised linoleic acid levels, suggesting an adaptive change in lipid metabolism under toxic stress.

Trichothecenes are known to disrupt enzymes that regulate fatty acid desaturation and elongation, thereby affecting membrane composition and signalling [15]. Increased levels of linoleic acid (C18:2n−6) may help maintain membrane fluidity or reduce oxidative stress. In insects, n-6 PUFA also serve as precursors for signalling molecules that govern development and moulting [2,3].

Nutritionally, higher linoleic acid improves the feed value of mealworm oil for monogastric animals and fish, which cannot produce this fatty acid [2]. However, an increased PUFA content may reduce oxidative stability, necessitating the use of antioxidants during storage. Similar changes in fatty acid profiles have been seen in other insects exposed to *Fusarium* toxins [24]. In vertebrates, such as Atlantic salmon, exposure to DON has been shown to lower plasma cholesterol and triglycerides [39]. This suggests a conserved effect of DON on lipid metabolism across different species.

From an applied perspective, the shift in fatty acid composition may not be harmful. An increased level of linoleic acid could enhance the potential of mealworm oil as a partial replacement for plant oils in aquafeeds. However, supplementation with long-chain n-3 polyunsaturated fatty acids (EPA, DHA) remains crucial [2,3].

In our previous study, conducted under the same experimental design, we demonstrated that *T. molitor* larvae tolerated DON-contaminated diets without adverse effects on survival or growth performance. We also observed that most of the ingested toxin was efficiently excreted in the frass, leading to only minimal accumulation in larval biomass [40]. The present study builds on these findings by offering mechanistic insights into DON metabolism in larvae, identifying multiple transformation pathways (sulfonation, glucuronidation, sulfation, glucosylation, and de-epoxidation), and linking these processes to noticeable changes in amino acid and fatty acid profiles. Together, these two complementary studies highlight a dual outcome: while *T. molitor* effectively prevents DON carry-over into larval biomass, thereby improving feed safety, detoxification is not metabolically neutral but involves nutritional shifts that may influence the quality of insect-derived protein and lipid fractions. This comparative perspective thus underscores both the safety potential and the nutritional trade-offs of using mealworms in sustainable feed production systems.

### 3.5. Ecological and Circular Economy Perspectives

The ability of *T. molitor* to metabolise DON has broader implications for food security and sustainability. Climate change is likely to increase *Fusarium* infections and DON contamination in cereals [16,17,18]. Therefore, strategies for safely utilising contaminated crops are essential. Insects offer a biological solution: by metabolising DON, they convert otherwise unusable feedstocks into high-quality protein and lipids, thereby supporting waste valorisation and circular economy models [2,6,41,42].

This approach also decreases dependence on traditional feed resources, such as soy and fishmeal, which have significant environmental impacts [1,4]. By enabling the safe use of contaminated substrates, insects can reduce food waste and enhance the resilience of feed supply chains against mycotoxin threats, supporting EU sustainability goals and circular bioeconomy strategies.

## 4. Conclusions

In summary, *T. molitor* larvae detoxify DON through multiple pathways, including sulfonation, glucuronidation, sulfation, glucosylation, and de-epoxidation, with significant contributions from gut microbiota. These processes reduce DON toxicity but involve metabolic costs, as indicated by changes in amino acid and fatty acid profiles. The dual outcome of detoxification and nutritional adjustment highlights the potential of insects as sustainable feed sources, while also emphasizing the need for combined assessments of feed safety, nutritional quality, and microbial roles. Additionally, the ecological role of insects in valuing contaminated biomass underscores their importance in future sustainable food systems.

### Key Contribution

*T. molitor* larvae metabolise DON through multiple pathways, such as sulfonation, glucuronidation, sulfation, glucosylation, and de-epoxidation, which reduces its toxicity.

DON exposure significantly modifies the amino acid profile of larvae, with notable reductions in essential amino acids (methionine, lysine, phenylalanine, and isoleucine) and key metabolic intermediates (aspartic and glutamic acids).

Extended exposure increases linoleic acid levels in larval fat, suggesting alterations in lipid metabolism associated with detoxification processes.

These results emphasise the dual role of *T. molitor* as a biological transformer of mycotoxins and a sustainable feed source. Our findings suggest that although DON undergoes detoxification, this process is accompanied by changes in the amino acid and fatty acid composition of the biomass. Therefore, feed safety assessments should consider not only residual toxin levels but also the nutritional quality of the material.

## 5. Materials and Methods

### 5.1. Feeding Substrates and Diet Preparation

The composition of the basal feed was analysed using a Foss InfraXact™ analyser (FOSS Poland, Warsaw, Poland).

To prepare diets A and B, DON (Sigma Aldrich, Poznań, Poland) was dissolved in methanol and sprayed onto the basal feed. The feed was thoroughly mixed and dried at approximately 50 °C under reduced pressure. Samples were collected from different locations within the batch to evaluate homogeneity and DON content using LC-MS/MS. The proximate compositions of the experimental diets are shown in Table 5.

Although near-infrared (NIR) spectroscopy (Foss InfraXact™) enables rapid, non-destructive assessment of feed composition, this method does not account for all nitrogen-containing compounds and specific carbohydrate fractions. As a result, the total of quantified nutrients does not reach 100%, and the remaining portion likely consists of undetectable minor constituents.

### 5.2. Feeding Experiment

Nineteen-week-old *Tenebrio molitor* larvae (average body mass: 112.47 ± 23.95 mg) were sourced from TENEBRIA Sp. z o.o. (Poland) and randomly divided into three dietary groups. Each group had six replicates, with about 50 g of larvae per replicate, housed in polypropylene containers (31 × 17 × 9.5 cm).

Group C: Control group fed basal diet;Group A: Fed DON-contaminated diet (663 µg/kg);Group B: Fed DON-contaminated diet (913 µg/kg).

Although the control diet (C) was prepared without intentionally adding DON, natural background contamination (28.7 µg/kg) was detected. This low level reflects unavoidable environmental contamination of cereal-based substrates and therefore prevented the use of a true negative control without DON. As a result, Diet C served as a practical control, representing the baseline contamination typically encountered in feed materials.

The DON concentrations in Diets A (663 µg/kg) and B (913 µg/kg) were chosen to represent low- and high-contamination scenarios. These values were selected based on levels frequently reported in naturally contaminated feeds in Europe [6,7,9,12], with Diet A approximating the lower range of field contamination and Diet B corresponding to higher but still realistic levels observed in cereals. This approach enabled us to model both moderate and severe dietary exposure conditions for larvae.

Larvae were raised for two weeks in a Memmert HPP 749 incubator (Memmert GmbH, Schwabach, Germany) at 27 ± 2 °C with 70 ± 5% relative humidity. Half of the replicates (*n* = 3 per group) were sampled after one week (C1, A1, B1), and the rest after two weeks (C2, A2, B2).

Unconsumed feed and frass were separated using 0.5 mm sieves. Larvae were starved for 24 h to empty their guts, then weighed and euthanised by freezing at −24 °C. During the trial, larvae were fed a mixture of 80% dry feed (control or DON-contaminated) and 20% fresh carrot pulp, which served as their water source. Feeding occurred once daily at 5:00 P.M. Mortality was monitored daily, while pupae and adult emergence were recorded weekly.

### 5.3. Determination of Deoxynivalenol, Its Metabolites, and Free Amino Acids Using HPLC-Q-TOF

#### 5.3.1. Extraction of Deoxynivalenol, Its Metabolites, and Free Amino Acids

Approximately 5 g of frozen larvae were ground using a ball mill (Millmix Agro, Madona, Latvia); amplitude 20, 30 s). One gram of homogenized larval biomass was transferred into 15 mL Falcon tubes, followed by the addition of 4 mL of solvent (methanol [VWR, Cat. No. 85800.320] (VWR International, Gdańsk, Poland) methanol–water 1:1, or water [VWR, Cat. No. 83645.320]), each containing 0.1% formic acid (Merck Sp. z o.o., Warsaw, Poland). Samples were shaken for 60 min using an Eberbach EL.680.Q.25 shaker (400 oscillations per minute) and centrifuged at 5000× *g* for 10 min at 4 °C (Thermo Scientific Sorvall X Pro Series, Thermo Scientific, Waltham, MA, USA). Supernatants were filtered through PTFE syringe filters (13 mm, 0.22 µm; VWR, Cat. No. 514-1275, VWR International, Gdańsk, Poland). A 980 µL aliquot of the filtrate was transferred to LC vials and spiked with 20 µL of enrofloxacin as the internal standard.

#### 5.3.2. Extraction Strategy for LC-QTOF Analysis

To optimize metabolite recovery, three extraction solvents were initially tested: (i) methanol, (ii) methanol–water (1:1, *v*/*v*), and (iii) water, each containing 0.1% formic acid. This approach was applied to account for the different polarities of target analytes, including DON, its metabolites, and free amino acids. Methanol was used to improve the recovery of less polar compounds, methanol–water mixtures enabled broad-spectrum extraction of moderately polar metabolites, while water favored highly polar analytes. After comparative evaluation, the extract showing the highest overall signal abundance and reproducibility in LC-QTOF analysis was selected for subsequent measurements. This strategy ensured efficient detection of DON-derived metabolites and amino acids while minimizing the risk of selective underestimation due to solvent polarity.

#### 5.3.3. LC–MS Reagents

The following LC–MS grade solvents and reagents were used: methanol, water, formic acid, and ammonium formate (Merck, Warsaw, Poland). All chemicals were purchased from commercial suppliers and used without further purification.

#### 5.3.4. Instrumentation–Chromatographic Conditions

Chromatographic analyses were performed using an Agilent 1260 Infinity III HPLC system (Agilent Technologies, Santa Clara, CA, USA) equipped with a binary pump (up to 600 bar), a thermostatted autosampler, and a column oven. Separation was achieved with an Agilent Technologies InfinityLabZorbax Eclipse Plus C18 Rapid Resolution HD 2.1 × 50 mm, 1.8-micron column (Part Number: 959757-902). The chromatography was carried out under these conditions: column temperature of 40 °C, flow rate of 0.4 mL/min, injection volume of 1 µL, mobile phase comprising channel A: H_2_O with 0.1% formic acid and 5 mM ammonium formate, and channel B: MeOH with 0.1% formic acid and 5 mM ammonium formate. The gradient program is detailed in Table 6. Total analysis time was 27 min.

#### 5.3.5. Instrumentation–LC–MS Conditions

The HPLC system was coupled to an Agilent Revident Q-TOF mass spectrometer (Agilent Technologies, Santa Clara, CA, USA) equipped with an Agilent JetStream Dual ESI ion source. The instrument was tuned in high-resolution mode with a mass range up to 1700 *m/z*. Mass spectrometry conditions are presented in Table 7.

#### 5.3.6. Data Acquisition Mode

Mass spectrometric data were gathered in All Ions mode over the 50–1000 *m/z* range at a rate of 6 spectra per second in both positive and negative ionisation modes. Three collision energies (0, 15, and 30 V) were used. Data were recorded in centroid mode.

#### 5.3.7. Software and Databases

Mass spectrometry data were processed using MassHunter Qualitative Analysis software (Version 10, Agilent Technologies, Santa Clara, CA, USA). For targeted data mining, the Target/Suspect Screening workflow was applied, which uses the Find by Formula algorithm with the PCDL Mycotoxins PCDL B.07.00 database. For untargeted analyses, the Compound Discovery workflow was employed, based on the Molecular Feature Extraction algorithm. Metabolite identification was supported by comparison against the PCDL Metlin_Metabolites_AM_PCDL database.

Heatmaps were created to show the relative levels of DON and its metabolites across different treatment groups and timepoints. The colour gradient (green–yellow–red) indicates normalized signal intensity from LC-QTOF analysis, where green signifies low levels, yellow intermediate, and red high levels. White areas denote values below detection limits or non-detectable signals. This visual approach allows for comparison of relative differences between groups but does not convey actual concentration values.

### 5.4. Fatty Acid Profile Analysis by GC-FID

Fatty acids were analysed using a modified Folch lipid extraction, followed by methylation and gas chromatography. About 5 g of frozen larvae were ground in a ball mill (Millmix 20, Tehtnica, Slovenia; amplitude 20, 30 s). Four grams of the homogenized biomass were transferred into 50 mL Falcon tubes with 30 mL of chloroform (Chempur, Cat. No. 112344305; Chempur, Piekary Śląskie, Poland)–methanol (VWR, Cat. No. 200864.320; VWR International, Gdańsk, Poland) in a 2:1 (*v*/*v*) ratio. Extraction was carried out in two cycles, each involving 30 min of shaking (Eberbach EL.680.Q.25, 400 oscillations per minute) and 10 min of sonication (Ultron U504 bath, Dywity, Poland, at 90% power).

Filtrates were collected using Whatman grade 597½ filters (4–7 µm; Cytiva, Wilmington, DE, USA, Cat. No. 10311844) and evaporated under nitrogen at 60 °C. Lipid residues were stored overnight at 4 °C. For methylation, 10 mL of chloroform–methanol–sulfuric acid (100:100:1, *v*/*v*/*v*; Chempur, Cat. No. 115750002; Chempur, PiekaryŚląskie, Poland) was added. One milliliter of the mixture was sealed in glass ampoules and incubated at 100 °C for 2 h in a Wamed SUP 30 W oven (Wamed, Warsaw, Poland). After cooling, samples were diluted 20-fold in methanol before analysis.

#### 5.4.1. Chromatographic Conditions

Fatty acid methyl esters (FAMEs) were analysed using a Perkin Elmer Clarus 500 GC-FID system (Perkin Elmer, Waltham, MA, USA) equipped with a ZB-FAME capillary column (30 m × 0.25 µm × 0.25 mm; Phenomenex, Torrance, CA, USA). The chromatographic separation was carried out under the following conditions: injection volume, 1 µL; split ratio, 50:1; oven temperature, 240 °C; carrier gas, helium; and constant flow, 40 cm/s. Oven program: 80 °C (2 min), ramp at 10 °C/min to 140 °C, ramp at 2 °C/min to 190 °C, ramp at 30 °C/min to 260 °C (hold 2 min), detector: FID at 260 °C, total runtime: 37 min.

#### 5.4.2. Calibration and Identification

Quantification of fatty acids was performed using an external calibration curve prepared with pentadecanoic acid (C15:0; Sigma-Aldrich, Poznań, Poland, Cat. No. P6125). The calibration curve included five concentration levels (79.76, 99.70, 199.4, 398.8, and 797.6 µg/mL), each measured in triplicate. Concentrations of individual fatty acids were determined based on this calibration curve. For qualitative identification, retention times were compared with a certified FAME Mix standard (C4–C24; Sigma-Aldrich, Poznań, Poland, Cat. No. 18919), enabling the identification of individual fatty acids present in the samples.

### 5.5. Statistical Analysis

All data were tested for normality (Shapiro–Wilk test) and homogeneity of variances (Levene’s test) before analysis. Differences between dietary groups (Control, A: 663 µg/kg DON, B: 913 µg/kg DON) and timepoints (week 1 vs. week 2) were assessed using one-way analysis of variance (ANOVA). Tukey’s HSD test was used when ANOVA showed significant effects (*p* < 0.05). For non-parametric data, Kruskal–Wallis tests were performed, followed by Dunn’s post hoc test with the Bonferroni correction. Results are presented as mean ± SD. Statistical analyses were conducted with Statistica 13.3 (TIBCO Software Inc., Palo Alto, CA, USA) and GraphPad Prism 9 (GraphPad Software, San Diego, CA, USA). A *p*-value < 0.05 was considered statistically significant.

## Figures and Tables

**Figure 1 toxins-17-00478-f001:**
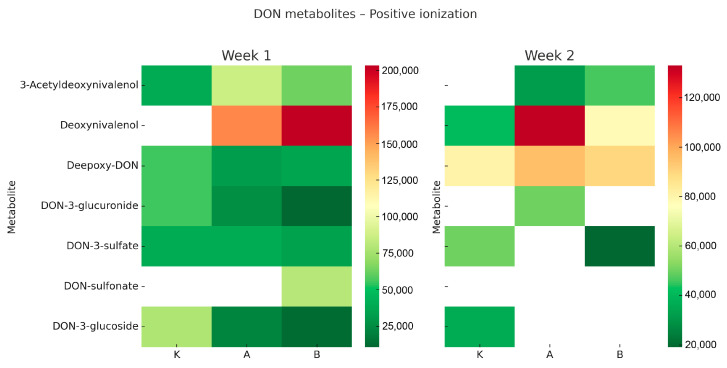
Heatmap showing deoxynivalenol (DON) and its metabolites in *Tenebrio molitor* larvae after one and two weeks of dietary exposure (positive ionisation mode). Group K—control, group A—low-dose DON, group B—high-dose DON. Relative abundance = normalised LC-QTOF signal intensity (arbitrary units). The colour scale (green–yellow–red) indicates low–medium–high metabolite levels.

**Figure 2 toxins-17-00478-f002:**
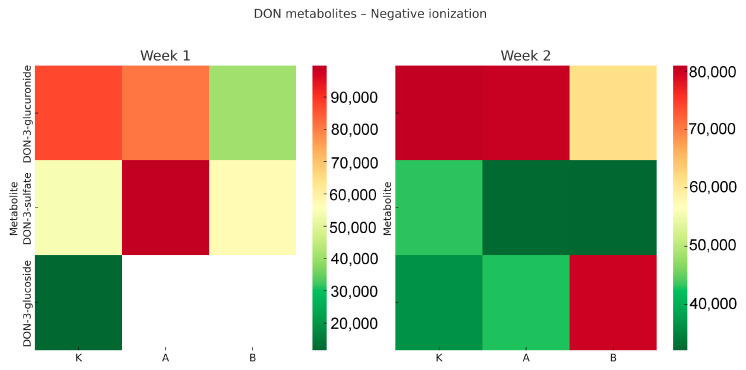
Heatmap showing deoxynivalenol (DON) and its metabolites in *Tenebrio molitor* larvae after one and two weeks of dietary exposure (negative ionisation mode). Group K—control, Group A—low-dose DON, Group B—high-dose DON. Relative abundance = normalised LC-QTOF signal intensity (arbitrary units). The colour scale (green–yellow–red) indicates low–medium–high metabolite levels.

**Figure 3 toxins-17-00478-f003:**
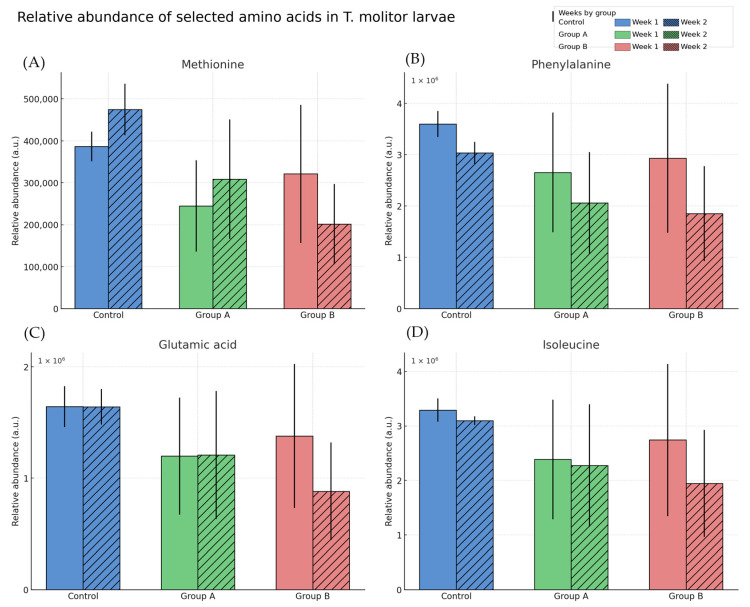
Relative abundance of selected amino acids in Tenebrio molitor larvae after one and two weeks of feeding on control (C) and DON-contaminated diets (Group A: 663 µg/kg, Group B: 913 µg/kg). Representative essential and non-essential amino acids are shown: (**A**) L-Methionine, (**B**) L-Phenylalanine, (**C**) L-Glutamic acid, and (**D**) L-Isoleucine. Values are expressed as mean ± SD. Relative abundance = normalised LC-QTOF signal intensity (arbitrary units).

**Table 1 toxins-17-00478-t001:** Relative abundance of free amino acids in Tenebrio molitor larvae after one week of feeding on control (C) or DON-contaminated diets (Group A; Group B). Values are presented as mean ± SD. Different superscript letters indicate statistically significant differences between groups (*p* < 0.05). Results show the relative abundance obtained through LC-QTOF analysis, not absolute concentrations.

Amino Acids	Week 1		
	Control (C)	Group A	Group B
L-Arginine	282,815.7 ± 16,013.5 ^a^	272,011.0 ± 24,435.1 ^a^	291,036.7 ± 10,274.3 ^a^
L-Aspartic Acid	243,678.7 ± 19,744.5 ^a^	189,926.3 ± 46,524.8 ^a^	240,021.0 ± 21,019.0 ^a^
L-Cysteine	13,703.3 ± 2523.5 ^a^	13,496.3 ± 4140.8 ^a^	12,687.0 ± 4234.3 ^a^
L-Glutamic acid	1,642,210.7 ± 184,031.7 ^a^	1,381,604.0 ± 458,771.5 ^a^	1,654,420.3 ± 271,522.8 ^a^
L-Glutamine	400,164.0 ± 197,852.1 ^a^	533,476.3 ± 269,281.8 ^a^	291,587.7 ± 48,977.7 ^a^
L-Histidine	720,202.0 ± 12,770.8 ^a^	744,021.3 ± 67,631.7 ^a^	669,297.0 ± 35,320.7 ^a^
L-Isoleucine	3,291,067.7 ± 215,250.0 ^a^	2,845,289.3 ± 548,891.6 ^a^	3,361,560.0 ± 264,704.1 ^a^
L-Lysine	71,704.7 ± 4519.7 ^a^	60,250.3 ± 16,736.8 ^a^	69,602.7 ± 1934.8 ^a^
L-Methionine	386,367.7 ± 35,184.9 ^a^	287,492.3 ± 74,653.1 ^a^	394,162.7 ± 28,928.1 ^a^
L-Phenylalanine	3,596,554.7 ± 253,678.9 ^a^	3,072,111.3 ± 979,713.1 ^a^	3,568,435.7 ± 377,955.0 ^a^
L-Proline	86,156.7 ± 15,017.0 ^a^	58,263.3 ± 33,964.2 ^a^	66,164.3 ± 11,755.1 ^a^
L-Serine	92,762.0 ± 8590.2 ^a^	83,429.3 ± 2628.9 ^a^	87,795.7 ± 6812.9 ^a^
L-Threonine	272,263.7 ± 14,326.8 ^a^	219,948.0 ± 25,463.6 ^b^	242,936.3 ± 17,099.0 ^ab^
L-Tryptophan	1,697,254.3 ± 67,430.3 ^a^	1,767,340.7 ± 325,829.0 ^a^	1,989,960.3 ± 215,419.2 ^a^
L-Tyrosine	514,143.7 ± 160,899.7 ^a^	557,547.0 ± 99,607.2 ^a^	420,043.7 ± 25,973.0 ^a^

**Table 2 toxins-17-00478-t002:** Relative abundance of free amino acids in *Tenebrio molitor* larvae after two weeks of feeding on either control (C) or DON-contaminated diets (Group A: 663 µg/kg; Group B: 913 µg/kg).

Amino Acids	Week 2		
	Control (C)	Group A	Group B
L-Arginine	292,570.7 ± 13,152.1 ^a^	275,944.3 ± 20,840.5 ^a^	275,801.0 ± 9632.0 ^a^
L-Aspartic Acid	302,491.3 ± 46,072.7 ^a^	241,711.7 ± 32,895.8 ^ab^	154,897.7 ± 30,152.3 ^b^
L-Cysteine	5998.7 ± 1102.9 ^a^	10,241.0 (single value) ^a^	8597.5 ± 614.5 ^a^
L-Glutamic acid	1,639,666.3 ± 159,828.8 ^a^	1,455,465.7 ± 216,570.3 ^ab^	1,073,695.0 ± 111,784.5 ^b^
L-Glutamine	348,709.0 ± 65,978.2 ^a^	478,472.7 ± 156,535.5 ^a^	477,642.7 ± 108,813.7 ^a^
L-Histidine	586,522.0 ± 44,468.7 ^a^	618,370.7 ± 71,084.7 ^a^	590,327.3 ± 45,560.8 ^a^
L-Isoleucine	3,097,908.3 ± 79,801.1 ^a^	2,766,483.7 ± 299,800.2 ^ab^	2,381,444.7 ± 213,192.8 ^b^
L-Lysine	79,244.0 ± 11,694.9 ^a^	66,183.7 ± 15,088.6 ^a^	55,289.0 ± 11,283.7 ^a^
L-Methionine	474,866.0 ± 61,335.3 ^a^	368,437.7 ± 69,098.6 ^ab^	242,663.0 ± 37,060.1 ^b^
L-Phenylalanine	3,033,179.3 ± 214,963.4 ^a^	2,489,748.0 ± 345,590.3 ^ab^	2,257,815.7 ± 219,568.0 ^b^
L-Proline	60,228.7 ± 7738.7 ^a^	44,248.7 ± 9109.1 ^ab^	31,436.3 ± 9718.5 ^b^
L-Serine	107,283.0 ± 5227.1 ^a^	95,788.0 ± 4821.6 ^ab^	84,510.3 ± 6971.5 ^b^
L-Threonine	255,015.7 ± 15,768.7 ^a^	226,650.3 ± 22,599.6 ^ab^	201,217.0 ± 16,484.0 ^b^
L-Tryptophan	1,823,412.0 ± 269,837.6 ^a^	1,905,375.0 ± 262,081.5 ^a^	2,248,000.7 ± 220,528.6 ^a^
L-Tyrosine	350,860.3 ± 4588.8 ^a^	470,978.7 ± 134,063.6 ^a^	369,504.7 ± 117,102.9 ^a^

Values are presented as mean ± SD. Different superscript letters indicate statistically significant differences between groups (*p* < 0.05). These results show relative abundance values obtained through LC-QTOF analysis, not absolute concentrations. For L-cysteine in Group A, the compound was detectable above the quantification threshold in only one of three replicates; therefore, only a single value is reported, and no standard deviation could be calculated.

**Table 3 toxins-17-00478-t003:** Fatty acid composition (mg/g fresh weight, mean ± SD) of *Tenebrio molitor* larvae after one week of feeding on a control (C) diet and a DON-contaminated diet (Group A, Group B). Superscript letters indicate significant differences between groups (*p* < 0.05).

Fatty Acids (mg/g)	Week 1		
	Control (C)	Group A	Group B
C14:1 cis 9 (myristoleic acid)	3.828 ± 0.325 ^a^	3.655 ± 0.145 ^a^	3.688 ± 0.115 ^a^
C16:0 (palmitic acid)	16.416 ± 0.459 ^a^	16.221 ± 0.994 ^a^	16.799 ± 1.083 ^a^
C16:1 cis 9 (palmitoleic acid)	3.324 ± 0.516 ^a^	2.848 ± 0.201 ^a^	2.904 ± 0.047 ^a^
C18:0 (stearic acid)	4.278 ± 0.403 ^a^	4.243 ± 0.017 ^a^	4.264 ± 0.136 ^a^
C18:1 cis 9 (oleic acid)	44.044 ± 1.371 ^a^	44.267 ± 0.619 ^a^	44.805 ± 2.647 ^a^
C18:2 cis 9,12 (linoleic acid)	22.234 ± 0.511 ^a^	22.300 ± 0.879 ^a^	23.002 ± 1.029 ^a^

**Table 4 toxins-17-00478-t004:** Fatty acid composition (mg/g fresh weight, mean ± SD) of *Tenebrio molitor* larvae after two weeks of feeding on control (C) and DON-contaminated diets (Group A, Group B). Superscript letters indicate significant differences between groups (*p* < 0.05).

Fatty Acids (mg/g)	Week 2		
	Control (C)	Group A	Group B
C14:1 cis 9 (myristoleic acid)	3.603 ± 0.150 ^a^	3.888 ± 0.076 ^a^	3.821 ± 0.161 ^a^
C16:0 (palmitic acid)	15.083 ± 0.700 ^a^	15.860 ± 0.938 ^a^	15.707 ± 0.244 ^a^
C16:1 cis 9 (palmitoleic acid)	2.877 ± 0.305 ^a^	2.908 ± 0.154 ^a^	2.894 ± 0.168 ^a^
C18:0 (stearic acid)	4.541 ± 0.763 ^a^	3.880 ± 0.083 ^a^	4.089 ± 0.089 ^a^
C18:1 cis 9 (oleic acid)	41.03 ± 1.438 ^a^	42.446 ± 3.052 ^a^	42.151 ± 0.632 ^a^
C18:2 cis 9,12 (linoleic acid)	22.602 ± 0.318 ^b^	24.753 ± 0.729 ^a^	24.352 ± 0.910 ^a^

**Table 5 toxins-17-00478-t005:** Proximate composition of the experimental diets.

Ingredient	Content [%]		
	Diet C	Diet A	Diet B
Ash	6.64		
Crude fiber	3.99		
Starch	37.68		
Protein	23.64		
Fat	5.28		
Moisture	10.61		
	Mycotoxin concentration (μg/kg)		
Deoxynivalenol	28.71	663	913

**Table 6 toxins-17-00478-t006:** Gradient program used for chromatographic separation in LC–QTOF analysis. The table shows the proportion of mobile phases A (water with 0.1% formic acid and 5 mM ammonium formate) and B (methanol with 0.1% formic acid and 5 mM ammonium formate) over time.

Time (min)	A (%)	B (%)	Flow (mL/min)
0.00	95.0	5.0	0.400
1.00	95.0	5.0	0.400
15.00	5.0	95.0	0.400
21.00	5.0	95.0	0.400
21.01	95.0	5.0	0.400
27.00	95.0	5.0	0.400

**Table 7 toxins-17-00478-t007:** Mass spectrometry source and acquisition parameters used for LC–QTOF analysis. The table lists ion source conditions, fragmentor, skimmer, collision energies, and acquisition settings.

All Ions Acquisition Parameters	*m/z* range: 50–1000 Acquisition rate: 6.0 spectra/s Time per spectrum: 166.7 ms
Ion Source Parameters	Gas temperature: 250 °C Drying gas flow: 8 L/min Nebulizer pressure: 40 psi Sheath gas temperature: 350 °C Sheath gas flow: 12 L/min Capillary voltage: 4000 V Nozzle voltage: 0 V
Additional Parameters	Fragmentor: 120 V Skimmer: 65 V Oct 1 RF Vpp: 400 V Quad Amu: 148

## Data Availability

The original contributions presented in this study are included in the article and in the Appendix A. Further inquiries can be directed to the corresponding author.

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
