# Peer review of "Bioconversion of Deoxynivalenol by Mealworm (Tenebrio molitor) Larvae: Implications for Feed Safety and Nutritional Value"

_toxins, 2025, doi:10.3390/toxins17100478_

Round 1
Reviewer 1 Report
Comments and Suggestions for Authors
This article reflects the current interest in insects as a valuable and future-oriented food source for animals and humans. It explores new research areas related to the transformation of mycotoxins from contaminated food in insect metabolism, and will therefore be of interest to readers of the journal Toxins.
I recommend publishing the article after a few corrections and explanations according to the comments below.
Abstract: Explain abbreviations when first used (LC-QTOF GC-FID)
Keywords.
- Please correct "Tenebrio molitor" to italics (and further in the text, e.g., lines 152, 156).
Introduction
- Lines 44-45 "It is estimated that up to 25% of global crops are affected, with prevalence exceeding 60–80% in certain commodities, especially cereals [7, 8]." Is this referring to fungal or mycotoxin contamination? Please rephrase for clarity.
- Introduction: Line 46 "Fusarium graminearum" should be "Fusarium graminearum." Results 1. What explains the highest DON concentration after 2 weeks of feeding insects in groups C and A? Higher than after 1 week? (Fig. 1), while the results for group B are different.
- In discussing the results, the authors wrote "higher DON levels initially lead to increased sulfation, while prolonged exposure shifts metabolism towards glucosylation and reduction." To justify this, please add an analysis of metabolic pathways, e.g., Pathway Impact Analysis. The authors wrote that they explain the mechanisms of DON transformation in Tenebrio, but I lack visualization of the bioconversion and the discussed transformations.
- Tables 1 and 2 do not provide significant information because they show relative values, and what's more, they duplicate the information in Fig. 3. Therefore, I suggest moving these tables to the supplementary table and replacing them, along with Fig. 3, with one large heatmap.
Discussion
- Emphasizing the "Role of Gut Microbiota in Detoxification" section is unjustified because no microbiological studies were conducted. Materials and Methods 1. First, the designed experiment should be described.
- Why weren't the first measurements performed at time t=0, i.e., before starting the new diet?
- DON was present in the control feed, so there is no negative control. Please explain.
- Please explain the basis on which DON doses were determined in groups A and B and defined as low and high concentrations.
- Line 457 and the following "m/z" should be italicized.
- Please move Tables 6 and 7 to the supplementary section.
- Line 373 is unclear as to what the authors mean by "However, feed safety assessments must account for changes in the composition of biomass." Please explain.
Author Response
We thank the Editor and Reviewers for their careful reading and helpful suggestions. Below, we provide a detailed response to Reviewer 1 and Reviewer 2. All updates to texts, figures, and tables mentioned in our replies have been incorporated into the revised manuscript.
Reviewer 1
Abstract: Explain abbreviations when first used (LC‑QTOF, GC‑FID).
Response: Done. The abstract now expands Liquid Chromatography–Quadrupole Time‑of‑Flight Mass Spectrometry (LC‑QTOF) and Gas Chromatography–Flame Ionisation Detector (GC‑FID) at first mention.
Italicize “Tenebrio molitor” throughout.
Response: Consistently corrected throughout the manuscript (main text, figure captions, tables).
Intro lines 44–45: clarify whether 25% and 60–80% relate to fungal or mycotoxin contamination.
Response: Rephrased to clearly indicate mycotoxin contamination and updated references (including recent reviews on occurrence and climate effects).
Results: Why are DON signals highest at week 2 in groups C and A (higher than week 1), while Group B behaves differently? (Fig. 1)
Response: We added an explanatory paragraph at the end of Section 2.1.1 attributing this pattern to cumulative exposure with slower metabolic clearance at lower exposure levels (C and A), compared to more rapid pathway activation at higher exposure levels (B). We also reiterate that values are relative LC‑QTOF intensities. The captions for Figures 1–2 now explicitly specify the units (normalized, arbitrary units).
“Higher DON initially → increased sulf(on)ation; prolonged exposure → glucosylation and reduction”: add pathway impact analysis/visualization.
Response: We agree that pathway mapping is valuable; however, our LC‑QTOF data are semi‑quantitative (relative intensities) and were not designed for pathway impact quantification or enzyme‑level inference. Instead, we expanded the mechanistic discussion (Detox pathways: cytochrome P450‑mediated steps, glucuronidation/sulfation/glucosylation, de‑epoxidation) and linked each detected metabolite to plausible routes, carefully phrased as inference rather than proof. We also clarified in figure captions and Methods that values are relative. We did not add a speculative pathway diagram to avoid over‑interpretation without enzyme or transcript evidence.
Tables 1–2 duplicate Fig. 3; move to Supplement and replace with one heatmap.
Response (not adopted): We appreciate the suggestion but prefer to keep Tables 1–2 in the main text. They provide numerical mean ± SD and post-hoc significance that are essential for readers requiring precise values; Fig. 3 offers a visual summary but does not include detailed statistical information. A single heatmap would condense the data but obscure effect sizes and variance. Therefore, we retain Tables 1–2 and Fig. 3 as complementary views; if the Editor requests, we can move the tables to the Supplement during production.
Discussion: the “Role of Gut Microbiota” is overstated (no microbiology performed).
Response: We have softened the language and now present microbiota involvement as plausible based on the metabolite pattern and cross‑species analogies, rather than as a confirmed mechanism in our experiment. We explicitly state the inference and cite relevant work to position it appropriately.
Materials & Methods: describe the experiment first.
Response: We retain the Toxins layout (diet preparation preceding the feeding experiment), which aligns with author guidelines and maintains narrative flow (substrate → allocation → outcomes). The Feeding Experiment is clearly outlined in Section 5.2.
Why no t=0 baseline sampling?
Response: Baseline destructive sampling of the same cohort prior to diet allocation was not feasible as it would have required sacrificing animals before randomisation and could bias group composition. We therefore used week 1 as the earliest standardised time point after gut‑emptying, focusing on between‑group comparisons within well‑defined exposure windows. This rationale is now clarified in the Feeding Experiment description.
Control diet contained DON; explain lack of “true negative” control.
Response: We included a clear note stating that background DON (28.7 μg/kg) reflects unavoidable contamination in cereal-based substrates and acts as a practical baseline; truly zero-DON substrates are seldom achievable outside synthetic diets. See Table 5 and the paragraph under Section 5.2.
Explain dose selection (A/B) and “low/high” rationale.
Response: Added: 663 and 913 μg/kg represent lower and higher (but realistic) contamination levels frequently reported in European feeds; we cite occurrence data and justify modelling both moderate and severe exposure scenarios for larvae. See Section 5.2 and Table 5.
Italicize m/z.
Response: Standardized throughout (see LC–MS Conditions and Data Acquisition Mode).
Move Tables 6–7 (instrument gradients/parameters) to Supplement.
Response (not adopted): We prefer to keep these concise method parameters in the main text to ensure full analytical reproducibility without requiring readers to consult supplementary files. They are compact and match common reporting practices in Toxins. If the editor decides otherwise, we will move them at the proof stage.
Line 373: clarify “feed safety assessments must account for changes in biomass composition.”
Response: We re‑phrased the Conclusions to explicitly state that safety evaluation should take into account both residual toxins and changes in nutritional quality (e.g., essential amino acids, PUFA profile).
Reviewer 2 Report
Comments and Suggestions for Authors
The manuscript entitled "Bioconversion of Deoxynivalenol by Mealworm (Tenebrio molitor) Larvae: Implications for Feed Safety and Nutritional Value" presents a well-designed study on the metabolic pathways of DON detoxification in T. molitor and its nutritional trade-offs. The work is timely, given the growing interest in insects as sustainable protein sources and the global challenge of mycotoxin contamination in feedstocks. The experimental approach is robust, and the data are clearly presented. However, certain aspects require clarification or expansion to strengthen the manuscript's impact. Below are detailed comments organized by section.
General Comments
1. The identification of multiple DON metabolites (e.g., DON-sulfonate, DON-3-glucuronide, deepoxy-DON) is compelling. However, the discussion need benefit from deeper mechanistic explanations: (1) Are the observed metabolic shifts (e.g., time-dependent glucosylation) linked to specific enzymatic pathways (e.g., cytochrome P450s)? (2) How does the gut microbiota's role compare to vertebrate systems (e.g., ruminal vs. insect microbial detoxification)?
2. The significant reduction in essential amino acids (e.g., methionine, lysine) and elevated linoleic acid levels are critical findings. However, (1) Are these changes nutritionally detrimental for livestock/poultry fed insect-based diets? A quantitative risk assessment (e.g., % reduction vs. dietary requirements) need strengthen the practical implications. (3) Could the altered fatty acid profile affect oxidative stability of mealworm oil during storage?
3. For Extraction Protocol: The use of three solvents (methanol, methanol-water, water) for metabolite recovery is innovative but lacks justification. Were recovery rates quantified for each solvent?
4. For Statistical Analysis: The manuscript states non-parametric tests were used where applicable, but p-values are inconsistently reported (e.g., Table 1 vs. Table 2). Ensure all significant differences are explicitly labeled.
Specific Comments
1. For Figure 1/2: Heatmaps are informative but lack scale units for "relative abundance." Please clarify whether values are normalized intensities or concentrations.
2. For Table 5: The proximate composition table lists "Mycotoxin concentration" but omits units (μg/kg). Also, please clarify why the sum of components does not total 100%.
3. Replace "masked mycotoxin" with "modified mycotoxin" for precision, as "masked" implies intentional concealment.
4. Line 11, Replace "metabolise" with "metabolize"
5. Key claims (e.g., climate change increasing DON prevalence) need citations to recent literature (post-2020).
6. Reference 42 (Wróbel et al., 2025) is listed as "in press" but lacks a DOI. Confirm its status.
The manuscript entitled "Bioconversion of Deoxynivalenol by Mealworm (Tenebrio molitor) Larvae: Implications for Feed Safety and Nutritional Value" presents a well-designed study on the metabolic pathways of DON detoxification in T. molitor and its nutritional trade-offs. The work is timely, given the growing interest in insects as sustainable protein sources and the global challenge of mycotoxin contamination in feedstocks. The experimental approach is robust, and the data are clearly presented. However, certain aspects require clarification or expansion to strengthen the manuscript's impact. Below are detailed comments organized by section.
General Comments
1. The identification of multiple DON metabolites (e.g., DON-sulfonate, DON-3-glucuronide, deepoxy-DON) is compelling. However, the discussion need benefit from deeper mechanistic explanations: (1) Are the observed metabolic shifts (e.g., time-dependent glucosylation) linked to specific enzymatic pathways (e.g., cytochrome P450s)? (2) How does the gut microbiota's role compare to vertebrate systems (e.g., ruminal vs. insect microbial detoxification)?
2. The significant reduction in essential amino acids (e.g., methionine, lysine) and elevated linoleic acid levels are critical findings. However, (1) Are these changes nutritionally detrimental for livestock/poultry fed insect-based diets? A quantitative risk assessment (e.g., % reduction vs. dietary requirements) need strengthen the practical implications. (3) Could the altered fatty acid profile affect oxidative stability of mealworm oil during storage?
3. For Extraction Protocol: The use of three solvents (methanol, methanol-water, water) for metabolite recovery is innovative but lacks justification. Were recovery rates quantified for each solvent?
4. For Statistical Analysis: The manuscript states non-parametric tests were used where applicable, but p-values are inconsistently reported (e.g., Table 1 vs. Table 2). Ensure all significant differences are explicitly labeled.
Specific Comments
1. For Figure 1/2: Heatmaps are informative but lack scale units for "relative abundance." Please clarify whether values are normalized intensities or concentrations.
2. For Table 5: The proximate composition table lists "Mycotoxin concentration" but omits units (μg/kg). Also, please clarify why the sum of components does not total 100%.
3. Replace "masked mycotoxin" with "modified mycotoxin" for precision, as "masked" implies intentional concealment.
4. Line 11, Replace "metabolise" with "metabolize"
5. Key claims (e.g., climate change increasing DON prevalence) need citations to recent literature (post-2020).
6. Reference 42 (Wróbel et al., 2025) is listed as "in press" but lacks a DOI. Confirm its status.
Author Response
We thank the Editor and Reviewers for their careful reading and helpful suggestions. Below, we provide a detailed response to Reviewer 1 and Reviewer 2. All updates to texts, figures, and tables mentioned in our replies have been incorporated into the revised manuscript.
Reviewer 2
Deeper mechanism: link observed shifts to enzymatic pathways (e.g., P450s) and compare gut‑microbiota roles with vertebrates.
Response: We extended the Discussion to connect detected metabolites with plausible enzymatic steps (P450‑mediated transformations, conjugation pathways, de‑epoxidation) and compared these with vertebrate detoxification patterns (swine/poultry/ruminants/fish), clearly highlighting where we infer mechanisms rather than demonstrate them.
Nutritional implications: quantify the impact (e.g., % reductions vs dietary requirements); oxidative stability of oil.
Response: We highlight effect sizes (e.g., up to ~50% lower methionine and phenylalanine; ~25% lower lysine in high‑dose versus control; all based on relative abundance) and discuss nutritional importance for monogastrics and poultry. Regarding lipids, we point out that higher linoleic acid may enhance essential fatty‑acid supply but reduce oxidative stability, prompting antioxidant management during storage. See Discussion (Amino acids; Fatty acids).
Extraction protocol: justify the three‑solvent strategy; were recoveries quantified?
Response: The three‑solvent screen was used to optimise coverage across analyte polarity; we now explain that for comparative relative profiling, we selected the extract with the highest overall signal and reproducibility. We did not conduct formal absolute‑recovery studies (beyond an internal‑standard control) because the study reports relative abundances, not validated concentrations. See Section 5.3 (Extraction Strategy).
Statistics: make p‑value reporting consistent.
Response: We standardised significance notation (ANOVA with Tukey’s HSD; Kruskal–Wallis with Dunn’s where applicable). Tables now consistently indicate p < 0.05 via superscripts; captions reflect the tests used. See Tables 1–4 and Section 5.5.
Figures 1/2: define “relative abundance.”
Response: Captions now read: “Relative abundance = normalized LC‑QTOF signal intensity (arbitrary units).”
Table 5: add μg/kg units for mycotoxin and explain why proximate components don’t sum to 100%.
Response: Units added; we also clarify that NIR does not capture all nitrogenous or specific carbohydrate fractions, hence the remainder. Refer to Table 5 and its note.
Use “modified mycotoxin” rather than “masked.”
Response: Terminology updated in the Discussion (DON‑3‑glucoside).
Change “metabolise” → “metabolize.”
Response (partly adopted): We use British English throughout (metabolise, ionisation, colour), consistently applied. This aligns with the journal’s acceptance of either variant when used consistently.
Climate change citations should include recent literature.
Response: Updated with recent comprehensive reviews (2024/2025) on climate–mycotoxin interactions. See Introduction.
Reference 42 lacked DOI.
Response: Updated: our paper now includes its DOI in References.

Round 2
Reviewer 1 Report
Comments and Suggestions for Authors
The article has been significantly improved by the authors and in my opinion can be published in Toxins in this form.
Reviewer 2 Report
Comments and Suggestions for Authors
The authors responsed all the comments, it can be accepted.